# Essential Oils and Melatonin as Functional Ingredients in Dogs

**DOI:** 10.3390/ani12162089

**Published:** 2022-08-16

**Authors:** Domingo Ruiz-Cano, Ginés Sánchez-Carrasco, Amina El-Mihyaoui, Marino B. Arnao

**Affiliations:** 1Alinatur Pet Food, S.L., Pol. Ind. Saprelorca, Lorca, 30817 Murcia, Spain; 2Department of Plant Biology (Plant Physiology), Faculty of Biology, University of Murcia, 30100 Murcia, Spain

**Keywords:** dogs, essential oils, functional ingredients, improved health, melatonin, nutraceuticals, pet food, phytogenics

## Abstract

**Simple Summary:**

Phytogenics are plant-based compounds with beneficial actions in feed technology and/or animal health. These so-called plant secondary metabolites are very diverse and with wide possible applications in humans and animals. Among them, essential oils (EOs) are the most used in feed for livestock and pets. Lately, melatonin has acquired new and interesting applications in dogs. Recent studies using EOs and/or melatonin in dog feeding and their involvement in health aspects are presented.

**Abstract:**

The use of nutraceuticals or functional ingredients is increasingly widespread in human food; their use is also widespread in animal feed. These natural compounds generally come from plant materials and comprise a wide range of substances of a very diverse chemical nature. In animals, these compounds, so-called phytogenics, are used to obtain improvements in feed production/stability and also as functional components with repercussions on animal health. Along with polyphenols, isoprenoid compounds represent a family of substances with wide applications in therapy and pet nutrition. Essential oils (EOs) are a group of complex substances with fat-soluble nature that are widely used. Melatonin is an indolic amine present in all living with amphiphilic nature. In this work, we present a review of the most relevant phytogenics (polyphenol, isoprenoid, and alkaloid compounds), their characteristics, and possible uses as nutraceuticals in dogs, with special emphasis on EOs and their regulatory aspects, applied in foods and topically. Additionally, a presentation of the importance of the use of melatonin in dogs is developed, giving physiological and practical aspects about its use in dog feeding and also in topical application, with examples and future projections. This review points to the combination of EOs and melatonin in food supplements and in the topical application as an innovative product and shows excellent perspectives aimed at addressing dysfunctions in pets, such as the treatment of stress and anxiety, sleep disorders, alopecia, and hair growth problems, among others.

## 1. Introduction

Functional ingredients are increasingly used in diets to improve some aspects of health. Also called nutraceuticals, they are presented as complementary ingredients to diets that can help prevent possible diseases or dysfunctions. These functional ingredients are included in the diet as dietary supplements in the form of pills, capsules, liquids, or tablets, or they can also be included in foodstuffs as enrichers, reinforcing them [1,2]. Although we can find lists of functional ingredients according to their possible role in general health or specific dysfunctions, the most common is to classify them by their chemical structure or biosynthetic origin [3,4].

Table 1, Table 2 and Table 3 show detailed lists of the most widely used functional ingredients, mainly from plants, classified according to the chemical structure of active ingredients. We did not include those well-established functional ingredients in the food/feeds such as mono- and polyunsaturated fatty acids (MUFA, PUFA), vitamins (A, E, C, etc.), and the so-called prebiotics (polysaccharides, dietary fiber, short-chain fatty acids, etc.) and probiotics (live microorganisms) [4,5].

Polyphenols are a category of secondary plant metabolites that incorporates a large number and variety of compounds (Table 1) [6,7,8]. Such a variety of compounds incorporates a great variety of possible benefits, being able to find several beneficial effects for the same subclass and common effects in different classes and subclasses [9,10,11]. Of all the polyphenols studied as nutraceuticals or functional ingredients, we can highlight those that are better known as curcuminoids, betalains, and stilbenes, such as resveratrol, quinones as ubiquinol, and various flavonoids very studied, such as luteolin, genistein, hesperetin, quercetin, catechinsm and anthocyanidins (Table 1) [12,13].

Another group or class of functional ingredients of great interest are terpenes/terpenoids/isoprenoids (Table 2). These compounds of oily nature, but without being fatty acids, are widely distributed in the plant kingdom, constituting the first line of chemical defense for plants [14,15,16]. Monoterpenes and monoterpenoids are the main constituents of essential oils (EOs), which also contain, to a lesser extent, sesquiterpenoids and phenylpropanoids. Mono- and sesquiterpenoids have a huge variety of components, their most studied functions being antibacterial, antifungal, and antiviral, as well as antitumor and anti-inflammatory properties; their possible applications as neuro-protectors and as prebiotics have been recently studied. Triterpenes/triterpenoids are well known and used, especially phytosterols as cholesterol-lowering agents. Saponins and limonoids still have a wide field of study to cover, while steroid triterpenoids such as cardenolides, classified as cardiac glycosides, are widely applied in heart dysfunctions (Table 2) [12,17,18,19].

The list of known alkaloids is equally extensive, although in this case, their use as functional ingredients is much more restricted due to their more widespread action on the nervous system (Table 3). However, new functions of many alkaloids and new alkaloids with new properties are being discovered, which augur new functional applications [12,20,21]. We must highlight the notoriety of glucosinolates, nitrogenous compounds synthesized from amino acids with excellent health properties that, in recent years, have encouraged a large consumption of brassica vegetables such as broccoli, cauliflower, romanesco, cabbages, Brussel sprouts, Pak-choi, turnips, rutabaga, etc., which have the highest content of these interesting compounds that inhibit mitosis and can stimulate apoptosis in human tumor cells (Table 3) [22,23,24,25,26].

## 2. Essential Oils

Essential oils (EOs) are a complex mixture of oily volatile compounds generated by plants with eco-physiological functions related to the attraction or repulsion of insects and other herbivores. These oils are responsible for the distinctive aroma associated with individual plant species. EOs are generated in special plant structures (uni- or multicellular) such as glands, glandular hairs, papillae, generally named trichomes, and oil ducts. They are mainly found in leaves but also in fruits and flowers. The extraction of EOs, generally by the traditional technique of steam distillation and now by the modern supercritical carbon dioxide extraction technique, has very low yields (around 1%), generating highly concentrated volatile mixtures of EOs [16,27,28,29,30,31,32,33,34].

The composition of the EOs is diverse, depending on the plant species and chemo-types, which vary according to geographical and genetic parameters (chemical polymorphism) [35]. The bioactive compounds of EOs are terpenes and terpenoids, mainly mono-terpenoids (90%) and, in a small percentage, sesqui-, di- and triterpenoids; as very minority components appear some phenylpropanoids (see Table 1) and others (amino acids, polyketides, and sulfur compounds).

Generally, EOs are classified according to their molecular composition considering their main component. Thus, we find the different families of EOs and their denomination; we can cite an example: EOs mostly constituted by hydrocarbons with the ending *-ene* or *-ane* (α-pinene, limonene, menthane, carane); if they are alcohols, ending *-ol* (menthol, geraniol, farnesol), if they are aldehydes, ending *-al* (citral, neral), if they are ketones, ending *-one* (α-thujone, carvone, fenchone), if they are oxides, ending *-ole* (cineole, ascaridole) (Figure 1).

Table 4 shows some examples of EOs widely used as functional ingredients. This is a tiny sample of the number of EOs we can use. Although there are studies with some of the particular components of EOs (mainly terpenoids), most studies in therapy have been done with complete EOs, because it is very difficult to obtain pure samples of many of the components.

Regarding the possible beneficial actions of EOs, the successive scientific studies that are continually appearing generally come to give evidence of the extensive ethnopharmacological knowledge that has been accumulating throughout history and tradition, both from the Western, Asian, and African traditions. Table 1, Table 2 and Table 3 show, in general, many of the beneficial actions of functional ingredients. Among the properties of these ingredients of interest to the pet food sector, we can point out, from general actions for health valued as healthy, energetic, invigorating, restorative, anti-aging, etc., to more or less specific actions such as antibacterial, antifungal, antiviral, and antiparasitic. Additionally, ingredients with regulatory activities of metabolic functions related to cholesterol, triglycerides, glucose, ureides, etc. We can also mention the ingredients against pain, nausea, dizziness, hypertension, vasodilators, etc., and those with the protective systemic activity of the liver, kidneys, heart, lungs, and, also of urinary, circulatory, gastrointestinal, oral, and nasal systems. Of special interest are the compounds intended to regulate or improve mood and sleep, such as antidepressants, relaxants, anxiolytics and sedatives, and finally, those with an activating capacity of the immune system and with anticancer capabilities. Therefore, it is common to find in EOs specifications on health benefits such as analgesic, stimulant, narcotic, hyper-, hypotensive, bronchodilator, antimicrobial, anti-tumoral, vermicide, antimalarial, anticholinergic, cholagogue, emetic, cardiotonic, sympathetic, vasoconstrictor, etc. A whole presumed arsenal of natural compounds to improve and cope with the health dysfunctions of our pets. We must not forget that they are not drugs and, therefore, only proper and generally continued use could, presumably, alleviate some specific health dysfunctions [12].

According to the ECHA (European Chemicals Agency), EOs are defined as a volatile part of a natural product, which can be obtained by distillation, steam distillation, or pressing in the case of citrus fruits. It contains mostly volatile hydrocarbons. EOs are derived from various sections of plants and are “essential” in the sense that they carry a distinctive scent or essence of the plant. The European Federation of Essential Oils (EFEO) and the International Fragrance Association (IFRA) have published guidance for characterizing EOs. In feeds, EOs are studied, considered, and authorized by diverse European Food Safety Authority’s (EFSA) Panels such as Food Additives and Nutrient Sources added to Food, Additives and Products or Substances used in Animal Feed, which provides scientific advice on the safety and/or efficacy of additives and products or substances used in animal feed. These EFSA’s Panels, generally composed of renowned experts from Europe, evaluate their safety and/or efficacy for the target species, the user, the consumer of products of animal origin, and the environment. It also analyzes the efficacy of biological and chemical products/substances intended for deliberate use in animal feed. If EFSA’s opinion is favorable, the European Commission prepares a draft regulation to authorize the additive. This is then discussed and endorsed by the Member States represented in the Standing Committee on Plants, Animals, Food and Feed, Section Animal Nutrition [36]. Scientific opinions were reported in EFSA journals to be considered by the scientific community.

The components of EOs were studied and classified by EFSA, which provides an excellent database of food flavorings containing information on it [37], the FDA Office of Food Additive Safety (OFAS), and the Flavor and Extract Manufacturers Association of the United States (FEMA), founded in 1909, comprise flavor manufacturers, flavor users, flavor ingredient suppliers, and others with interest in the flavor industry, maintained available an online registered of flavorings, including EOs. Additionally, the International Organization of the Flavor Industry (IOFI), a global association representing the industry that creates, produces, and sells flavorings worldwide, provides excellent access to consult natural (complex) substance lists applied to food and feeds [38].

The EOs are more to be considered as flavorings. Generally, the EOs used in treating diseases in humans are also recommended for animals, with some exceptions, especially in cats. The use of EOs in animal/pet food requires, as in humans, some precautions and recommendations. According to the FDA, “GRAS” is an acronym for “Generally Recognized As Safe”. This term is applied to EOs. Most EOs and their components are classified as GRAS. FEMA, through its Food Additives Committee, has been publishing since 1960 multiple reports providing valuable information on EOs, giving their average dose of application and their maximum limits of use in various foods. Furthermore, since the publication of the first edition in 1971, Fenaroli’s Guide, a handbook that remains the standard reference for flavor ingredients worldwide, including GRAS substances recognized by FEMA and FDA [39]. In short, a Panel’s evaluation of an EOs or other substance performs a toxicity study observation, its relevance to observed effects on human/animal health is evaluated, and the dose at which no adverse effects (NOAEL) are observed is determined. NOAEL values are often used as reference points in the calculation of margins of safety (MOS) by taking the ratio between the NOAEL and the estimated intake for the substance under consideration. NOAEL values have commonly been used as a point of reference for the calculation of MOS, acceptable daily intake, and similar values by regulatory bodies. Mathematical modeling of the dose–response data was applied to estimate the benchmark dose corresponding to a specific change in response compared to the background, increasing relevance in safety evaluations [40].

According to the EC Regulation on additives for use in animal nutrition (EC1831/2003, Annex I), EOs could be included in different categories considering their possible action. Therefore, the added EOs can be considered a sensory additive if it exerts an improvement in the smell or palatability of the feed and also as a zootechnical additive, improving feed digestibility and/or gut microflora, but also as physiological stabilizers because the EOs can favorably affect animal health, for example by improving stress tolerance or safeguarding against possible infections. Although EOs can also be considered as technological additives, specifically as preservatives and antioxidants, protecting feed against deterioration by microorganism and oxidation [41,42]. In the United States, pet food products do not require premarket approval by the FDA.

Phytogenics is the term applied to plant-based compounds with some positive effects on animal growth and health. These phytogenics can occur in various forms, such as herbs, spices, raw plants and their extracts, oleoresins, EOs, vegetable oils, and hydrolates [43,44]. Very often, the format is not specified, being a very important feature to know its composition, characteristics, and possible beneficial possibilities in animal nutrition. In many cases, the lack of specificity in the type of phytogenic used has led to results largely inconsistent with limited understanding. Thus, to conduct systematic and comprehensive evaluations of the efficacy and safety of phytogenic compounds due to their complex composition, a better classification of phytogenics in animal nutrition studies may be necessary [43]. In addition to herbs and spices, widely used in feeds, EOs have been widely studied in animal nutrition, especially since restrictions on the use of antibiotics appeared. Studies on the bactericidal, fungicidal, and viricidal activities of many EOs are numerous, especially in pigs and poultry [45]. Also in dogs, there are numerous studies on EOs and their application as possible sepsis controllers [46,47,48,49,50].

Table 5 shows some examples of the use of EOs in dogs, indicating the dosage and possible beneficial effect. Generally, EOs have been used in dogs to improve their general health and immunological response, but also in specific therapies such as those aimed at improving liver, renal, cardiovascular, gastrointestinal, muscular-joint, and skin health, among others.

Although most of the applications of EOs are their topical use to avoid or reduce infestations and/or dermal affections [61], in some cases, EOs were included in food supplements, that is, as a nutritional supplement to the dog’s usual diet. Due to the lipophilic nature of the EOs, it is necessary to apply a fat matrix for perfect solubilization of the EOs, such as salmon oil. In this case, in addition to the beneficial qualities of salmon oil (richness in omega-3) and its natural content in tocopherols and carotenoids, the EOs come to complement their antioxidant properties. Increasing the intake of certain functional ingredients (vitamin D, omega-3 PUFA, phytogenic such as some essential oils and tea catechins) positively affects immunological function, improving defenses and reducing the risk of infection [67]. One of these products has recently been marketed, containing a range of EOs with functional specificities such as antiparasitic action, with savory, sagebrush, and clove EOs, or joint-articular improvement, with eucalyptus, ginger, and marjoram EOs [68].

Although EOs are classified as safe substances (GRAS) for use in animal nutrition, as discussed above, we cannot stress enough that aspects of animal food safety are crucial when using EOs. Gossypol toxicity is a well-known example. The intensive search for more affordable protein sources in fish nutrition in aquaculture led to the use of cottonseed protein concentrate, a product with excellent nutritional qualities, except for the existence of gossypol, a sesquiterpene aldehyde, toxic at accumulated high levels. The existence of gossypol causes intestinal inflammatory problems, as has been demonstrated in Nile tilapias, grass carps, and turbots, also in cattle acts as a cardiac and reproductive toxicant [69]. Perhaps, only ruminant microflora appears to have the ability to inactivate the presence of gossypol in feed [70].

## 3. Melatonin

Melatonin, an indoleamine synthesized from tryptophan and with multiple functions in mammals, was discovered in 1958 in cows [71] and in 1959 in humans [72,73]. This molecule, considered the sleep hormone, received this name due to its first studied action, its aggregator effect of melanocytes in frogs, tadpoles, and some fish, but not in mammals [71]. Melatonin is structurally *N*-acetyl-5-methoxytryptamine, a derivative of serotonin (5-hydroxytryptamine) that acts as a hormone on multiple endogenous rhythms such as temperature, mood, immune system, and other hormones and metabolic pathways [74,75,76,77]. Melatonin is secreted by the pineal gland to the cerebrospinal fluid and then to the bloodstream, with maximality during the middle of the night. The best-known action mechanism of melatonin is its regulatory role in sleep–wake cycles. Melatonin acts as a signal of darkness, providing information to the brain and other organs, synchronizing endocrine rhythms. Melatonin adjusts the timing of oscillator elements of the central and peripheral biological clocks. For this reason, melatonin is often recommended to relieve the symptoms of jet lag, a disorder in sleep rhythms due to transoceanic or long-haul flights crossing multiple time zones [78,79,80].

Melatonin is also an excellent antioxidant, presenting an antioxidant capacity several times greater than classical antioxidant molecules such as ascorbic acid, glutathione (GSH), and vitamin E. The scavenging activity of melatonin against reactive oxygen species (ROS) and reactive nitrogen species (RNS) has been clearly demonstrated by Reiter’s group and others, both in vitro and in vivo assays. This antioxidant action of melatonin has been reinforced by an antioxidative cascade that occurs due to the generation of some of its reaction products such as cyclic-3-hydroxymelatonin, N_1_-acetyl-N_2_-formyl-5-methoxykynuramine (AFMK) and N_1_-acetyl-5-methoxykynuramine (AMK), in vitro and in vivo detected [81]. In addition, melatonin activates the expression of antioxidant enzymes in animal tissues in response to oxidative stress and exposure to toxic agents. Thus, superoxide dismutases (SOD), catalases, GSH peroxidases, GSH transferases, GSH synthases, and others are clearly up-regulated by melatonin to cope with possible distress situations. Generally, melatonin can regulate ROS/RNS levels by (i) a direct chemical action, scavenging its, in a receptor-independent action, and (ii) regulating antioxidant enzyme expression in a receptor-dependent action to control the redox network [82].

Although the possible beneficial actions of melatonin in diseases of mammals are very diverse (Figure 2) [83], relevant beneficial effects have been verified in studies on neurological dysfunctions (chronic fatigue, age-thermoregulation, migraine, multiple sclerosis, fibromyalgia, depression, schizophrenia, etc.), sleep disorders (jet-lag, insomnia, delayed sleep-phase, night-shift work sleep, sleep quality in blindness and autism), and others such as gastrointestinal, cardiovascular, in pre- and post-operative stress. Currently, research focuses on its role as an adjuvant in radio- and chemotherapies against several types of cancer, its role in neuro-dysfunctions such as Alzheimer’s and Parkinson’s, in diabetes and metabolic syndrome, and in immunological diseases, and its role as a relevant anti-inflammatory agent, having been proposed as a therapy against SARS-CoV-2 [74,75,84,85,86,87,88,89,90,91,92,93,94,95].

In dogs, melatonin has been used in various therapies. The use of melatonin is authorized, and it is the veterinarian who must prescribe its treatment in case of canine diseases or dysfunctions. Melatonin is usually given to the pet in the form of pills, powder, drops, or gelatin capsules, although it is often added to the feed in the form of a powder or liquid. It is also possible to apply it in creams or lotions in the case of skin treatments. There are at least three applications of interest related to melatonin. These are: (1) seasonal (recurrent) alopecia of the flanks; (2) anxiety and nervousness about noise or disorientation; and (3) sleep disorders in elderly dogs.

### 3.1. Seasonal (Recurrent) Alopecia of the Flanks

Cyclic flank alopecia is a follicular dysplasia characterized by altered coat quality, as well as alopecia that affects the sides and trunk [96]. As the skin is exposed, dry skin and hyperpigmentation frequently occur. It was thought to be due to seasonal changes, but it has been proven that, although seasonal changes can influence, it does not have to be related. It usually occurs cyclically or recurrently with loss of follicles and is of unknown etiology. Generally, the skin has no itching or inflammation. It usually occurs between autumn and winter. The breeds in which this process has been described the most are Boxer, Doberman, Rhodesian, Schnauzer, French and English Bulldog, and some types of Terriers such as Airedale. However, it can also appear in their crosses. A certain predisposition in females has also been described. Moreover, although initially it was believed that this alopecia was triggered by some hormonal imbalance, it has been shown that this is not the cause [96,97,98,99,100,101,102,103].

Melatonin intake seems to help in the regrowth of new hair, invigorating the hair follicles and giving rise to a renewed and healthy coat. In some recurrent breeds, melatonin is usually given as a preventive treatment a few weeks before the date on which alopecia occurred the previous year. In a Boxer dog study, the supply of 6 mg of melatonin per day induced partial hair regrowth in 2 months and its complete regeneration at 4 months [104]. Studies have also been described in which melatonin treatment has not been successful against alopecia, possibly due to a defect in dose or other causes [105,106]. In other dermatological assays in dogs, unequal results in melatonin treatments have been described [107].

A similar dysfunction, so-called Alopecia X (also known as an adrenal hyperplasia-like syndrome, hyperadrenocorticism, or Cushing’s syndrome), is characterized by partial to complete alopecia of the neck, tail, dorsum, perineum, thighs, and trunk. In addition, the skin may become hyperpigmented, primarily in areas of alopecia, occurring in both female and male young and adult dogs.

Behrend and Kennis (2010) review the possible hormonal aspects that affect Cushing’s syndrome (hyperadrenocorticism). With respect to melatonin, data and arguments for and against the syndrome were presented [108]. Thus, as data in favor, a trial with 29 dogs with Alopecia X in melatonin treatments at doses between 3 and 6 mg/kg, every 12 h for 4 months, where 15 of the patients showed hair regrowth [109]. Against in the same study, treating dogs with melatonin and mitotane (a toxic inhibitor of steroid hormone production that results in a decrease in cortisol levels), partial or complete hair regrowth was observed in only 62% of all dogs. Increasing the dose of melatonin in 8 dogs only showed hair regrowth in one of them. Of the five dogs whose hair did not grow back while receiving melatonin, two exhibited complete hair regrowth, one partial hair regrowth, and two were unable to regrow hair while receiving mitotane [109]. Additionally, in 15 Pomeranians with Alopecia X, melatonin (1.0–1.7 mg/kg, twice a day) for 3 months, only 40% had mild to moderate hair regrowth [110].

Mammalian skin is not only a target of melatonin bioactivity but also an important site of biosynthesis, regulation, and metabolism, where melatonin was detected in hair follicles as a modulator of hair growth and/or pigmentation [111,112], also as an antiaging cream treatment [113]. The scarce data on dogs are supported by previous studies on humans [114]. In a randomized controlled trial, women with androgenetic alopecia treated topically with daily 0.1% melatonin solutions for 6 months had a significant induction of the anagen phase of hair [115], not without criticism [116]. Additionally, in an open-label study of women and men with androgenetic alopecia, melatonin (33 ppm) topical application reduced the alopecia degree, and the TrichoScan technique showed a 29.2% increase in the hair count in 54.8% of patients after 3 months, and a 42.7% increase in 58.1% of patients after 6 months, demonstrating improvements in hair texture, decreased hair loss, and a reduction in seborrheic dermatitis, in safety and tolerant topical application of a cosmetic melatonin solution [117]. An exhaustive and recent revision of melatonin action in melanocyte physiology in humans and other mammals can be consulted [118].

In this regard, our preliminary data show that topical melatonin plus EOs application in dogs with Leishmaniosis showed a significant improvement in the affectation, decreasing the area of dermatitis, skin itching, and therefore, erosion wounds gradually improved in treatments with melatonin + EOs cream every 2 days, for 4–5 weeks (Figure 3).

### 3.2. Anxiety, Phobia, and Nervousness Due to Noise or Disorientation

Melatonin behaves as an anxiolytic molecule with relevant sedative and calming properties and is used to prevent or treat states of anxiety or depression [119,120,121,122,123,124,125,126], also in menopausal disorders [127]. In the surgical pre-operative, dogs treated with melatonin required lower doses of the anesthetic propofol due to the previous calming action of melatonin [128]. Additionally, similarly, the efficacy of pre-operative sedation with melatonin to reduce intraoperative use of midazolam (a benzodiazepine) in women under total abdominal hysterectomy [129]. As in humans, stress and distress are common problems in pets, and more so when they are alone at home. Taking advantage of the properties of melatonin as a mild sedative, veterinarians may recommend its use in situations of nervousness or phobia. Dogs tend to suffer from various forms of anxiety attacks. Some have noise phobias and get anxious due to thunderstorms, alarms, sirens, vacuum cleaners, and others. Other dogs show signs of anxiety when they must be put in a car. For others, anxiety appears with separation from the owner and being left alone. Many veterinarians treat nervous dogs with melatonin or sedative herbs before they experience the cause of anxiety, inhibiting it. It is advisable to administer a dose of melatonin at least half an hour before anxiety-provoking events occur. Recently, a direct relationship between the relaxing properties of medicinal plants such as valerian roots and phytomelatonin content has been described [130].

### 3.3. Sleep Disorders in Elderly Dogs

There are multiple pieces of evidence that melatonin deficiency in the blood tends to cause insomnia, especially in older dogs, as it also occurs in humans [131,132,133,134]. The use of melatonin as a nutraceutical in elderly dogs to treat insomnia and anxiety has been proposed [135,136]. Previously, Aronson (1999) suggested the use of melatonin with amitriptyline to manage thunderstorm phobia [137]. Additionally, Fourtillan et al., 2002 presented a study on beagle dogs with melatonin synthetic analogs suggesting that insomnia may be treated by administering hypnotic acetyl metabolites of melatonin or their synthetic analogs [138]. Some authors have recommended melatonin doses of 3–9 mg for dogs and 1.5–6 mg for cats to improve cognitive dysfunction syndrome, sleep-wake disturbances, and anxiety in older pets [139,140,141]. In a clinical study with 14 dogs on sleep behavior disorders, melatonin treatment showed uneven results. Thirteen of the dogs did not show any improvement in insomnia when treated with melatonin, but neither did they with gabapentin, diazepam, diphenhydramine, acepromazine, clonazepam, or phenobarbital. Only one dog had a positive response when 3 mg of melatonin (at night) was added to the potassium bromide treatment, and another dog was reported to have no response to an unknown dose of melatonin. Probably, according to the authors, dosing problems or frequency could explain these results [142]. Zanghi et al. (2016) suggested that in older dogs, the decrease in locomotor activity and insomnia could be related to a gradual degenerative neuronal disconnection between dorsal, dorso-medial, and ventral subparaventricular zones within the paraventral hypothalamic nucleus that regulates circadian melatonin release [143]. Additionally, in a comparative study, Lefman and Prittie (2019) proposed future studies to improve psychogenic stress in hospitalized veterinary patients treated with diazepam, midazolam, alprazolam, lorazepam, dexmedetomidine, trazodone, and melatonin, through therapies less aggressive and healthier [144]. Recently, Bódizs et al. (2020) presented a review on sleep in dogs, which includes the possible roles of melatonin [145].

In addition, the melatonin treatments can be of great help in elderly dogs or even in those that, due to vision problems, are unable to distinguish between daylight and darkness due to the beneficial action of melatonin on the retinal level [146,147].

Other applications of melatonin in dogs are under study. For example, extrapineal melatonin can act as a gastrointestinal protector [148,149,150], in oral surgery and implant dentistry, increasing bone-to-implant contact values and new bone formation [151,152,153], as regenerative in the skin [59,118], also as canine anti-tumoral agent [154,155,156,157].

Regarding possible side effects of melatonin in dogs, very few effects have been described if it is administered correctly and with the appropriate dose. In fact, it is the lack of secondary effects which often makes it a better choice than tranquilizers or other drugs [158]. However, there are some side effects that should be considered and your veterinarian should be informed about them, as they may recommend a lower dose or a different treatment. Some side effects that may occur are upset stomach and cramps, tachycardia, itching, and confusion.

## 4. Conclusions and Future Perspectives

In this review, an exhaustive list of phytogenics, their classification according to their chemical structure, and some of their beneficial properties were presented. Particular mention is made of essential oils, analyzing their composition and safety aspects, such as nutraceuticals for application in dogs. The most interesting cases and their uneven results were analyzed. As a synergistic therapy, results of the action of melatonin in dogs and also in humans were presented for its application in dysfunctions of interest. The combination of EOs and melatonin is presented as an innovative product of possible topical or oral administration, with excellent properties aimed at dysfunctions of great interest such as the treatment of stress and anxiety, sleep disorders, alopecia and hair growth problems, vision dysfunctions related to advanced age, improvements in the immune and gastrointestinal system, and antitumor actions, among others. Both components have excellent compatibility in terms of solubility, complemented by their respective antioxidant properties, ensuring the high stability of functional solutions [68,82,159,160,161,162].

However, the application of EOs and/or melatonin in pet foods presents some challenges to consider, such as (1) the correct dosage based on safety parameters, where the recommendations of the different authorized agencies (EFSA, FDA) are crucial; (2) ensuring the stability of components against industrial pet food techniques such as extrusion, sterilization, etc., is an important task considering wet or dry pet food formulae; (3) the functionality of the preparations involves important in vitro and in vivo scientific studies; (4) the innovative techniques of nanoencapsulation is presented as a tool to improve functionality and stability; (5) the preferential use of natural components over synthetic substances even with the handicap of cost, e.g., natural EOs versus analogous synthetic substances, and phytomelatonin (of plant origin) versus synthetic melatonin [163]; and (6) adequate information on the pet food labels in order to convey to consumers the most interesting aspects related to the functional properties of food supplements and their components.

## Figures and Tables

**Figure 1 animals-12-02089-f001:**
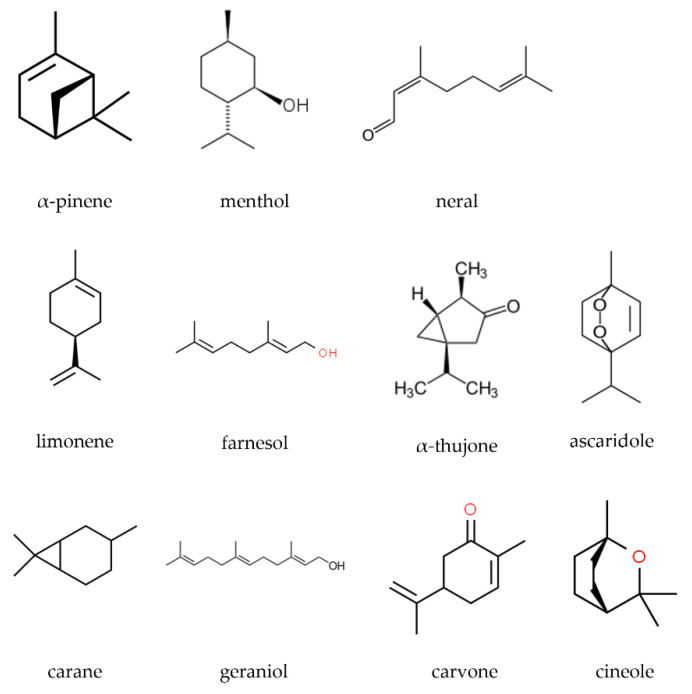
Molecular structures of some EO components.

**Figure 2 animals-12-02089-f002:**
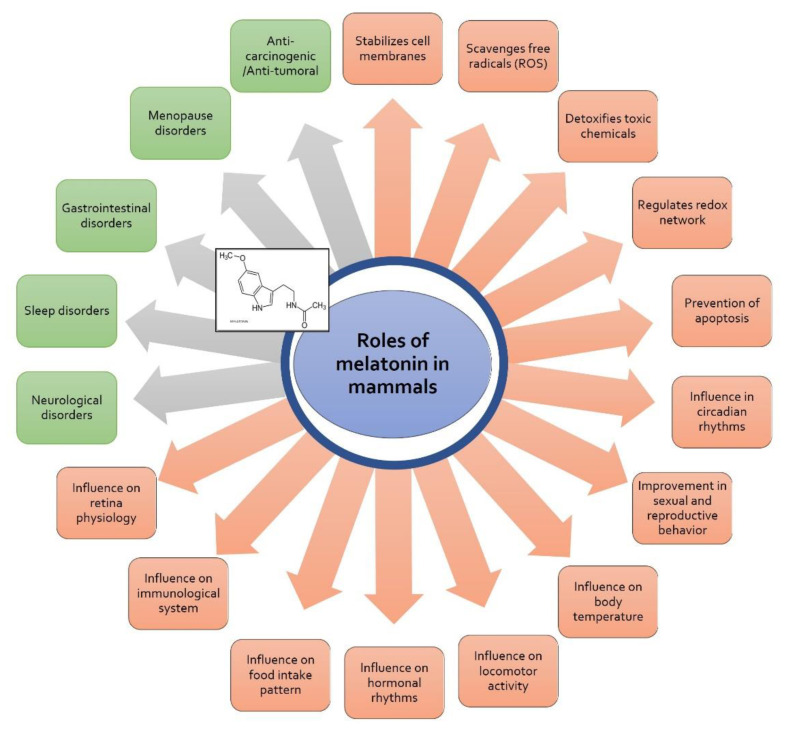
Roles of melatonin in mammals at cellular/physiological level. Their use in several disorders is shown in green color.

**Figure 3 animals-12-02089-f003:**
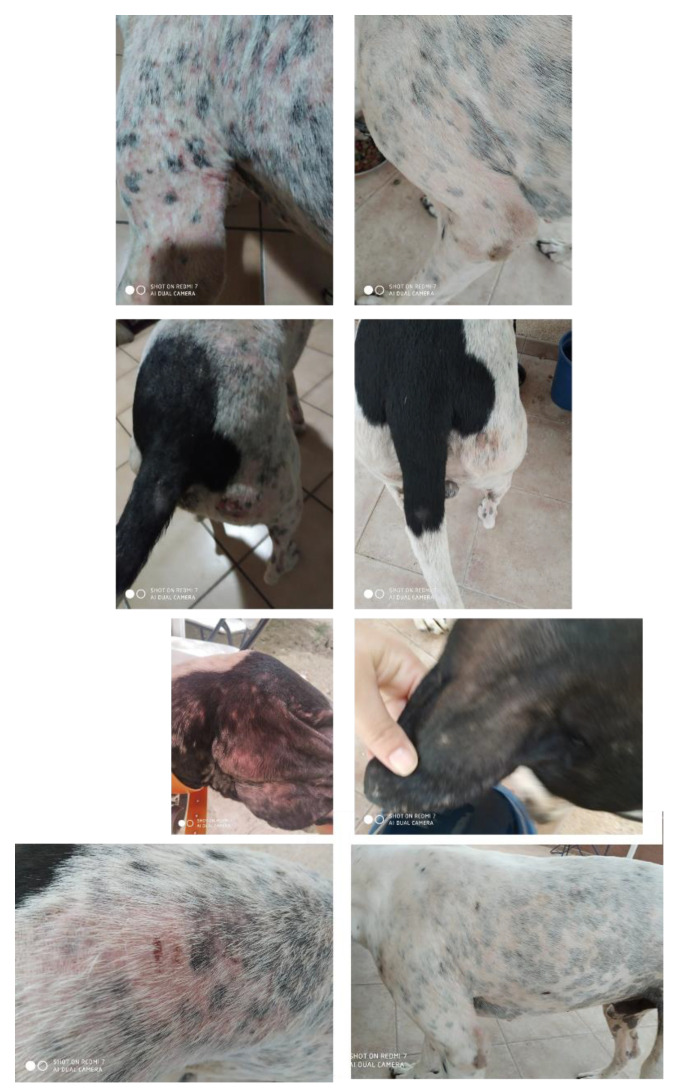
Photographs of different areas of the animal showing different affectations before the treatment (photos on the **left**) and 24 days after the treatment with 11 topical applications (photos on the **right**).

**Table 1 animals-12-02089-t001:** Natural functional ingredients and potential benefits: polyphenols.

Chemical Category/Class	Chemical Name/Subclass	Example of Compounds	Potential Benefits
**Phenylpropanoids**			
	*Simple phenols*	Arbutin, tyrosol	Antiseptic, diuretic, anti-tumoral
	*Hydroxycinnamic acids* *Free forms* *Esters* *Alcohols, Aldehydes* *& Glycosides*	Ferulic, caffeic, cinnamic Chlorogenic, rosmarinic, cynarin, cichoric, caftaric Coniferyl, caffeoyl, feruloyl, vanillin, eugenol	Antioxidant, chemoprotector, immunomodulatory, neuroprotector, dyspepsia, hypercholesterolemia
	*Acetophenones*	Apocynin, androsin, piceol, picein	Antiasthmatic, anti-inflammatory, neuroprotective, sedative
	*Salicylates*	Salicin, salicortin, populin	Analgesic, febrifuges, sciatica, myalgia
	*Curcuminoids*	Curcumin, dimethoxy- and bisdemethoxy-curcumin, and breakdown metabolites	Anti-inflammatory, anti-tumoral, cardioprotective, wound healing, anti-arthritis, antioxidant, anti-depressive
	*Lignans & Neolignans*	Pinoresinol, masoprocol, silybin, schizandrin, podophyllotoxin, enterodiol	Hypoglycemic, chemoprotector, antioxidant, keratosis, antifungal, anti-inflammatory, anti-tumoral, phytoestrogen precursors
	*Coumarins & Furanocoumarins*	Coumarin, aesculetin, xanthotoxin, umbelliferone, psoralen, angelican, bergapten, khellin	Photosensitizer, anti-vitiligo, psoriasis, tinea hypopigmentation, spasmolytic, bronchodilator, asthma, anti-hypertensive, renal calculi, hay fever, rhinitis
	*Betalains* *Betacyanins* *Betaxanthins*	Betanin, (iso-, pro-, neo-) Vulga-xanthin (mira-, portula-, indica-)	Antioxidant, antimicrobial, anti-tumoral
**Polyketide-derived**			
	*Stilbenes*	Resveratrol, pinosylvin, piceatannol, piceid, pallidol, viniferin, pterostylbene	Anti-inflammatory, neuroprotective, anti-tumoral, cardioprotective, anti-aging, antioxidant, antifungal, hypoglycemic
	*Quinones* *Naphthoquinones, Naphthodiantrones, Anthraquinones & Kavalactones*	Ubiquinol (Q10), menaquinone (vit K), plastoquinone, phylloquinone Juglone, lapachol, plumbagone, shikonin, hypericin, sennosides, carmine, fagopyrin, emodins, rhein, kavain, yangonin, methysticin	Anti-tumoral, anti-leukemic, antimicrobial, antiparasitic, antifungal, antiviral, anti-inflammatory, cardioprotective, laxative, hypnotic, sedative, anesthetic
**Flavonoids**			
	*Flavones*	Apigenin, luteolin, baicalein	
	*Isoflavones*	Genistein, diadzein, biochanin	
	*Flavonones*	Naringenin, eriodictyol, hesperetin, liquiritin	Antioxidant, anti-tumoral, anti-microbial, antiviral, anti-atheromatous, anti-hypertensive, anti-inflammatory, hepatoprotective, endothelial protection, cardioprotective, neuroprotective, chemoprotective, immunoprotective, estrogen-mediated responses, anti-aging
	*Flavonols*	Quercetin, kaempferol, myricetin, isorhamnetin	
	*Flavanols*	Catechin, epicatechin	
	*Flavan-3-ol (OPC)* ^1^	Epicatechin-3-gallate, epigallocatechin-3-gallate	
	*Anthocyanidins*	Malvadin, cyanidin, delphinidin, europinidin, pelargonidin, peonidin, rosinidin, aurantinidin	
	*Tannins* *Gallo- & Ellagitannins* *Condensed tannins (Proanthocyanidins)*	Galloyl derivatives, ellagic acid, punicalagin, rugosin-D, oenthein-B, sanguiin, geraniin, agrimoniin, puncialin, corilagin Procyanidins (OPC), propelargonidins, prodelphinidins, profisetinidins, proteracacinidins, theaflavins	Anti-tumoral, anti-inflammatory, antioxidant, antidiarrhoeic, anti-hemorrhagic, antimicrobial, hypolipidaemic, astringent, sclerosis, cardioprotective, endothelial function, platelet function, anti-hypertensive, anti-atherosclerotic, oral health

^1^ OPC—oligomeric procyanidins.

**Table 2 animals-12-02089-t002:** Natural functional ingredients and potential benefits: Terpenes and terpenoids.

Chemical Category/Class	Chemical Name/Subclass	Example of Compounds	Potential Benefits
**Monoterpenes/oids** **(main constituent of essential oils)**	*Regular* *Monocyclics* *Acyclics* *Bicyclics* *Irregular* *Iridoids* *Pyrethrins* *Cannabinoids*	Limonene, terpineol, menthol, thymol, p-cymene, carvacrol Linalool, citronelle, geranial Camphor, α-pinene, thujone Nepetalactone, valtrate, harpagide, oleuropein Pyrethrin, chrysanthemic acid, cinerin, jazmolone Δ^9^-tetrahydrocannabinol, cannabidiol, cannabicylol	Antifungal, antibacterial, antioxidant, anticancer, anti-spasmodic, analgesic, vasodilator, cardiovascular protector, anti-inflammatory, antidiabetic, anti-obesity, gut microbiota modulator, sedative, hepatoprotector, chloleretic, laxative, antiviral, insecticidal Euphoriant, analgesic, neuroprotective, antiemetic, anxiolytic, anti-tumoral, anti-inflammatory, bronchodilator
**Sesquiterpenes/oids**	*In EOs* *Lactones*	Bisabolol and its oxides, matricin, chamazulene, gossypol, zingerbene Germacrene, achillin, artemisin, cnicin, parthenolide, tanacetin, helenalin	Anti-inflammatory, wound-healing, contraceptive, anesthetic, antibacterial, antifungal, anti-protozoal, analgesic, anti-tumoral
**Diterpenes/oids**	*Acyclic, mono-, bi-,* *tri-, and tetracyclic*	Forskolin, marrubiin, paclitaxel, andrographolide, ginkgolides, bilobide, stevioside, rebaudioside, abietic acid, hautriwaic acid	Antihypertensive, vasodilatory, bronchodilatory, platelet aggregation inhibition, anti-tumoral, intraocular pressure regulator, hepatoprotector, immunomodulatory, neuroprotection, anti-diabetic, sweetener
**Triterpenes/oids**	*Free Phytosterols* *Limonoids*	Lanosterol, ganosterol, lupeol Sitosterol, campesterol, gugusterol, stigmasterol, brassicasterol, avenasterol, cycloartenol Azadirachtin, limonin, nomilin	Blood cholesterol and LDL level regulator, hypocholesterolemic, hypolipidemic, anti-obesity, cardio-, neuro-, thyroid-protective, anti-tumoral Antifeedant, insecticidal
	*Saponins* *Non-steroidal* *Steroidal* *(so-called Cardenolides/Bufanolides, including some* *cardiac glycosides *)*	Glycyrrhicin, ginsenosides, jujubosides, asiatoside, betulin Diosgenin, sarasapogenin, ruscogenins, withaferin-A Digitoxin *, digoxin *, convallatoxin *, cimarin *, proscillaridin *	Many systemic effects: antiallergic, anti-tumoral, immunomodulatory, anti-(bacterial, fungal, viral), cardio-, hepato-, neuro-protective, hypoglycemic, estrogenic-, digestive-regulator, hypocholesterolaemic, hearth arrhythmia & failure *, angiogenesis inhibitor *, apoptotic *, autophagic *, neuroprotective *
**Tetraterpenes/oids**	*Carotenes* *Xanthophylls* *Gukulenins*	α-, β-, γ-, and δ-Carotene, lycopene, phytoene Lutein, xanthins (viola-, luteo-, zea-, β-crypto-, astha-, anthera, cantha-), crocetins, and crocins Gukulenin A and B	Antitumoral, pro-vitamin A, hypocholesterolemic, cardiovascular protection, neuroprotector, immunoactivator, skin protection Antitumoral
**Meroterpenes**	*Terpenophenols*	Bakuchiol, ferruginol, totarol, epiconicol	Antioxidant, antibacterial, anti-inflammatory, ocular protection

* indicates cardiac glycosides.

**Table 3 animals-12-02089-t003:** Natural functional ingredients and potential benefits: Alkaloids and glucosinolates.

Chemical Category/Class	Chemical Name/Subclass	Example of Compounds	Potential Benefits
**Alkaloids**	*From lysine* *From ornithine* *From tryptophan* *From phenylalanine/tyrosine* *Steroidal (alkaloid saponins)*	Lupanine, cytosine, sedamine Cocaine, hyoscyamine, nicotine Vincamine, yohimbine, physostigmine, ergotamine, quinine, camptothecin Ephedrine, berberine, emetine, morphine, capsaicin, eserpine, ergotamine, caffeine Solanine, veratrine, solasodine	Analgesic, stimulant, narcotic, hyper-, hypotensive, bronchodilator, antimicrobial, anti-tumoral, vermicide, antimalarial, anticholinergic, cholagogue, emetic, cardiotonic, sympathetic, vasoconstrictor, antiasthmatic, anthelmintic
**Glucosinolates & derivatives**	*Aliphatic* *Aromatic* *Indolic* *Sulfur-derivatives*	Glucoraphanin, sinigrin Gluconasturtiin, glucotropaelin Glucobrassicins Isothiocyanates (allyl, benzyl), sulforaphane	Cancer prevention, anti-tumoral, antibacterial, antifungal, antioxidant, bronchodilator, skin irritation shooting

**Table 4 animals-12-02089-t004:** Components of some useful essential oils as functional ingredients of different plants.

Common Name	Scientific Name	Compounds *
Anise	*Pimpinella anisum*	**trans-Anethole, γ-**himachalene, estragole, 2-methyl-isoeugenol, anisaldehyde
Basil	*Ocimum basilicum*	**Linalool**, **1,8-cineole, methyl eugenol**, estragole, myrcene
Bergamot	*Citrus bergamia*	**Limonene, linalyl acetate,****γ-terpinene, linalool,** β-pinene, β-bisabolene
Cinnamon	*Cinnamomum zeylanicum*	**Eugenol,** β-caryophyllene, benzyl benzoate, cinnamyl acetate, α-phellandrene
Chinese tea tree	*Malaleuca alternifolia*	**Terpinen-4-ol, γ-terpinene, α-terpinene, 1,8-cineole, α-terpineol,** p-cymene, terpinolene, α-pinene
Clove	*Syzygium aromaticum*	**Eugenol, β-caryophyllene,** α-humulene, δ-cadinene
Eucalypt	*Eucalyptus globulus*	**1,8-Cineole**, **α-pinene,** limonene, p-cymene
Fennel	*Foeniculum vulgare*	**Anethole, fenchone,** α-pinene, limonene, estragole, anisaldehyde, β-phellandrene
Ginger	*Zingiber officinale*	**Geranial, neral,** geraniol, limonene
Hypericum	*Hypericum perforatum*	**α-Pinene, β-caryophyllene,** methyl-2-octane, dodecanol, myrcene
Lavender	*Lavandula angustifolia*	**Linalyl acetate, linalool, terpinen-4-ol, ocimene,** 1,8-cineole, limonene, camphor
Lemongrass	*Cymbopogon citratus*	**Geranial, neral,** geraniol, geranyl acetate, β-caryophyllene
Marjoram	*Thymus mastichina*	**1,8-Cineole, linalool,** α-terpineol, α-pinene, limonene, linalyle acetate
Peppermint	*Mentha piperita*	**Menthol, menthone,** 1,8-cineole, menthylacetate, isomenthone, neomenthol, menthofurane, limonene, β-caryophyllene
Rosemary	*Rosmarinus officinalis*	**α-Thuyone, α-pinene, camphene, camphor,** limonene, myrcene, borneol
Sagebrush	*Artemisia vulgaris*	**α-Thuyone, lyratol, 1,8-cineole,** camphor, β-thuyone, artemisinin
Salvia	*Salvia officinalis*	**α-Thuyone, camphor, 1,8-cineole, α-humulene,** β-thuyone, α-pinene, bornyle acetate, limonene
Savory	*Satureja montana*	**Carvacrol,****p-cymene,** γ-terpinene, thymol
Thyme	*Thymus vulgaris*	**1,8-cineole, β-phellandrene, camphor,** α-pinene, myrcene, borneol, limonene, neral

* In bold, the majority components.

**Table 5 animals-12-02089-t005:** Some recent examples of EO use in dogs and its benefits.

Plant/EOs/Dose/App Form	Animals	Benefits	Refs.
Lavender/0.18 mL/inner pinnas of both ears	Beagles	↓ sympathovagal activity ↑ relax and calming	[51]
*Artemisia absinthium* in vitro bioassay	Dogs	↑ acaricidal activity ↓ egg and larvae of *Rhipicephalus sanguineus dog* tick	[52]
Menthol and thymol oils applied as gel	Adult dogs	↓ buccal halitosis	[53]
Thymol and eugenol EOs 10 mL/kg, applied all over the skin and hair	English cocker spaniel dogs	↓ larvae of *Rhipicephalus sanguineus dog* tick	[54]
Turmeric oil at 2.5% in spray	Dogs with tick infestation	↓ number of tick bites In vitro effectivity: turmeric > DEET > PMD	[55]
Otogen^®^, EOs (tea tree, thyme, sage, eucalyptus, rosemary, lavender), and vegetable oil (macadamia and sunflower) 7 days applied	Dogs of different breeds and ages	↓ external otitis ↓ head shaking, erythema, and scraping	[56]
Thymol, cinnamaldehyde, and carvacrol; also clove and oregano EOs In vitro assay	Dogs (bacterial and *Malassezia pachydermatis* isolates)	↑ bactericidal and fungicidal activity ↑ Gram-positive bactericidal activity ↓ canine otitis	[57]
Dermoscent BIO BALM^®^ Neem, rosemary, lavender, clove, tea tree, oregano, peppermint EOs, cedar bark extract, and PUFAs Topical administration (0.6–2.4 mL weekly)	Dogs with low, medium, and severe atopic dermatitis	↓ canine atopic dermatitis and pruritus score ↑ beneficial in ameliorating the clinical signs of atopic dermatitis	[58]
Dermoscent BIO BALM^®^ Topical administration	Dogs of different breeds	↓ canine idiopathic noncomplicated nasal hyperkeratosis	[59]
Dog food containing EOs (clove, rosemary, and oregano; also, vit. E) vs. synthetic antioxidant BHT	Dogs of different breeds and ages	↓ lymphocytes, fecal bacterial count, oxidative stress (ROS), ↑ NPSH and glutathione S-transferases, feed conservation	[60]
Microencapsulated thymol, carvacrol, and cinnamaldehyde300 mg/kg of feed	Beagle dogs	neutrophils, lymphocytes, globulins, nitrogen oxide, GSH-POX ↓ ROS, fecal bacterial count, *Salmonella*, *Escherichia coli*	[44]
Cinnamon, thyme, clove, geranium, and tea tree EOs; also, eugenol, geraniol, cinnamaldehyde, thymol, and carvacrol individual components In vitro assay	Dog and human skin fungal dermatosis	↑ fungicidal activity, higher in dermatomycetes ↑ anti-mycosis therapy	[61]
Citrus, basilic, eucalyptus, cinnamon, lemon balm, lemongrass, lemon verbena, tea tree, savory, myrrh, and cannabis EOs	Possible application to dogs with pyoderma	↑ bactericidal activity against methicillin-resistant *Staphylococcus* ↑ pyoderma therapy	[62]
*Vernonia brasiliana* EO (Asteraceae)	Antileishmanial activity against *L. infantum* promastigotes and cytotoxicity on canine DH82 cells	↑ Antiparasitic effect, ROS, cell death by apoptosis ↓ mitochondrial membrane potential Antagonistic interaction with miltefosine drug	[63]
*Schinus molle* EO (Anacardiaceae)	Acaricidal effect on females and larval stages of *R. sanguineus*	EO (2%) caused larval mortality (99.3%) Inhibition of oviposition, egg hatching, and reproductive efficiency	[64]
*Tagetes minuta* EO (Asteraceae)	Acaricidal effect in vitro and on dogs of *R. sanguineus*	100% efficacy against larvae, nymphs, and adults of the tick on all tested conditions	[65]
Thyme and oregano EOs	Bacterial and fungal isolates from canine otitis externa	EOs showed good in vitro bactericidal and fungicidal activity against 100 isolates from dogs with otitis externa, including some highly drug-resistant isolates	[50]
Cinnamon EO	*Staphylococcus* strains from canine otitis	Effective antimicrobial and antibiofilm activity Potential alternative to treat ear infections in canines	[66]

↑ indicates increase activities, ↓ indicates decrease activities.

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
