# Peer review of "Essential Oils and Melatonin as Functional Ingredients in Dogs"

_animals, 2022, doi:10.3390/ani12162089_

Round 1

Reviewer 1 Report

It is very interesting because it was very well organized about phytogenic functional foods for dogs.So I judged to be very good as a nutritional review.

Since there are some parts of the English expression that make me feel a little "dumb", I will ask you to proofread the English again.

I've determined that the content is sufficient for publication in this magazine.

Reviewer 2 Report

The paper is a review of the effects of supplementation EOs and melatonin in dogs. This is an unusal combination, but is explained the end that the end of the manuscript (line 407-413) of which the authors in a previous report have presented this as an innovative product. 

The review of the effects EOs and melatonin is good, however the above mentioned point and the conclusion in the abstract (line 29-30) is far-fetched and not scientifically sound. Melatonin supplmentation needs veterinary prescription, EOs does not. The effects of EOs are multiple (Table 5), melatonin has clinical effect in some diseases and metal disorders in dogs. 

Suggestion for title:  Essential oils and melatonin as functional ingredient in dog food

   The conclusions need to be considered carefully.

Reviewer 3 Report

It appears that a thesaurus-type program may have been used.  Some word choices are not correct for the context (incorporated vs included for example).

This paper may be best redone as several smaller scope papers with more details.

Tables 1-3: Potential benefits are very vague and are not referenced.  Are these benefits all from oral ingestion via food?  Or what route is needed? Adverse effects are not mentioned.

Page 8, aromatherapy:  Have these been shown to be beneficial in pets? Which ones?  What dosing regimen?  What concentration?

Page 9-10: Animals is a worldwide journal, any other regulations (US, Australia, etc.)?

Page 10:  Is there a concentration limit on these oils to be considered GRAS?

Table 5: Doses/concentrations? In vitro assays are not strong science.  What were the adverse reactions?

Page 13: Melatonin section seems to be unrelated to the rest of the paper.  This could be a paper on its own.

Page 14: How are pets chewing gum?

Page 15: More shiny and effective way? Please be more specific.

Page 15: Alopecia X - the way this section is written it appears that melatonin decreased hair growth.

Page 17: Multiple pieces of evidence - please list them

Page 17: Please expand the common about challenges in future perspectives.  How do you dose in food?  How do you make it consistent?  What about degradation during storage?  Or when exposed to room air?

Round 2

Reviewer 3 Report

Title: The title does not fit with the contents of the paper.  There is no discussion about using these compounds in food, only that they may have certain benefits.  Potential benefits of phytochemicals in pets would be more fitting for the contents.

Abstract: The statements in the abstract are not supported by the paper.  It does not demonstrate innovative products or any use of these compounds in dog food.  There is some mention of using them as nutraceuticals but nothing pertaining to being part of the diet (how much would be needed, how does it remain stable, how do you ensure the safety of the product).  Please rewrite the abstract to fit the contents of the paper.

Table 1: Are all of these potential benefits in dogs? I cannot find this within your reference list.  If you are putting these agents forward as beneficial in a species it would be helpful to use those references.  If these are in other species (mice, rats) or in vitro, it should be noted as such.  We know that many compounds fail when used in vivo due to many reasons (bioavailability, species differences in metabolism, etc.).  Are these benefits seen with oral ingestion?  If you are promoting these compounds for use in food, they need to be beneficial by that route.

Table 2: See comments for table 1.   Also under monoterpenes the irregular label appears to be misplaced.  Same with Triterpenes Steroidal - there is also a close parenthesis missing (or an extra open parenthesis)

Table 3: See comments for table 1.

Page 9: Lines 136-157 - This paragraph seems out of place.  It appears to be more of a summary and is inserted within two paragraphs that describe production of EOs.

Page 10: Line 186-187 - This statement is not true.  There are many EOs that are not used in pets (esp cats) due to their differences in metabolism and sensitivity to adverse reactions.

Page 10: Lines 189-204 - Do these approved GRAS levels have any efficacy against disease?  Aren't NOAELs by definition so low there are not any effects at all?

Page 11: Line 228-229 - Please reference some oral studies with EOs in dogs showing sepsis control.

Table 5: The paper is titled Functional Ingredients in Pet Food, but these are mostly not oral studies (only one is). These studies do not support your claims.

Reference 45: This appears to be a self-reference and lay summary of this paper you have submitted.

Page 13: Line 259-260 - Gossypol is a well known cardiac and reproductive toxicant in cattle.

Page 13: Melatonin - This section is completely different than the rest of the paper that talks about plant based products.  This should be a separate paper.

Page 15: Line 321-322 - Reference please

Page 15: Line 332-333 - Reference

Page 15: Line 332-340 - Do we know if it is better than placebo?  Isn't 2-4 months a normal hair regrowth cycle?

Page 15: Line 346 - Remove 'In an interesting' from this sentence

Page 15: Lines 347-358 - These sentences need rewritten for clarity. Which dogs received what therapies? Who got the increased dose of melatonin?

Page 16: Lines 375-384: Topical application in humans. Can this be extrapolated to dogs?  Don't dogs groom themselves?  Would this oral ingestion change the kinetics?

Page 16: Lines 386-389 - This study with dermal application is another reason why the melatonin section should be in its own paper.  This does not demonstrate use in pet food.

Page 18: Line 418 - This sentence is repeated below.  All of these references (126-129) are in humans.  What is your dog reference?

Page 18: Line 423-444 - Did any of these studies show benefits? Also anxiety treatments for dogs have progressed greatly since 1999.

Page 18: Line 449 - It appears we are missing part of a paragraph.  This starts with 'Therefore' but the preceding section does not mention vision.

Conclusion: First paragraph - This is not supported by the body of the paper.  At no time do you discuss the use of this combination for health benefits.  There are no supporting studies for most of these claims. 

Conflicts of Interest - With your self-referencing of a product on the market (four times in fact), I believe their may be a conflict.

Author Response

see file
